# A Geo-Spatial Analysis for Characterising Urban Sprawl Patterns in the Batticaloa Municipal Council, Sri Lanka

**Mathanraj Seevarethnam** [1,*] **, Noradila Rusli** [2] **, Gabriel Hoh Teck Ling** [1] **and Ismail Said** [3]

1   Department of Urban and Regional Planning, Faculty of Built Environment and Surveying,
    Universiti Teknologi Malaysia, Skudai, Johor Bahru 81310, Malaysia; gabriel.ling@utm.my
2   Centre for Innovative Planning and Development (CIPD), Department of Urban and Regional Planning,
    Faculty of Built Environment and Surveying, Universiti Teknologi Malaysia, Skudai,
    Johor Bahru 81310, Malaysia; noradila@utm.my
3   Department of Landscape Architecture, Faculty of Built Environment and Surveying,
    Universiti Teknologi Malaysia, Skudai, Johor Bahru 81310, Malaysia; b-ismail@utm.my
*   Correspondence: mathanrajs@esn.ac.lk

**Abstract:** Urban sprawl related to rapid urbanisation in developed and developing nations affects sustainable land use. In Sri Lanka, urban areas have mostly expanded in a rather spontaneous, unplanned manner (based on the current settlers' subjective movement) rather than conforming to the local government's development plan. This growth inevitably leads to uncontrolled urban sprawl in many Sri Lankan cities, including Batticaloa. So far, Sri Lanka's planners or researchers have not yet tackled the sprawling developments in this city. Understanding the different forms and patterns of urban sprawl is the key to address sprawling growth. This study aims to identify the characteristics of urban sprawl in the Batticaloa municipal council using Geographic Information System (GIS) and remote sensing technology. Landsat satellite images for the years 2000, 2010, and 2020 as well as 2002, 2011, and 2019 population data were used and analysed using ArcGIS' maximum likelihood classification tool and the density function, respectively, to delineate the characteristics of urban sprawl. The results revealed that low-density development, leapfrog development, commercial ribbon development, and scattered growth are the influencing characteristics of urban sprawl in the Batticaloa municipality. These characteristics were found mainly in the urban edge of the city and have led to urban sprawl. The finding provides knowledge into recognising the characteristics of urban sprawl with empirical evidence. It affords a clear direction for future studies of urban sprawl in rapidly growing cities that are numerous in Sri Lanka, and the identified characteristics of urban sprawl can be useful in minimising future sprawl. This result can be a tool for future urban planning and management in the Batticaloa municipality.

**Keywords:** urban sprawl; land use; urbanisation; leapfrog development; scattered development

## 1. Introduction

Urbanisation is a reflection of the human activities affecting the land that has been threatened by the enormous pressure from population growth [1]. Rapid urban growth is generally related to and driven by the concentration of population in an area [2]. According to the United Nations' world population prospects in 2019, there will be an increase in the next 30 years of two billion people, from the current world population of 7.7 billion to 9.7 billion in 2050. Further, this increase will grow to almost 400 cities in the early 21st century, which includes around 70% in developing countries [3], including Sri Lanka. Currently, the urban population of Sri Lanka is almost 25% of the total population, which is expected to increase by 65% by 2030, which will cause cities to grow physically and numerically, creating urban sprawl issues in the future [4].

Urban sprawl has attracted much attention among policymakers and researchers in developed [5,6] and developing [4,7–9] countries worldwide. Most arguments for

urban sprawl are not based on strong empirical evidence but rather on speculation and assumptions [10]. Many researchers explain this concept with the urban environment of the research area. So far, there is no consensus on the definition of urban sprawl. Further, urban sprawl is a socioeconomic phenomenon that has gradually become a critical issue in many urban areas [11], including Sri Lankan cities. Built-up area, which is often considered a parameter for measuring urban sprawl [2], especially settlement density or size, can only afford an overall measure of the urban form [12]. Increasing urban sprawl results in household preferences, the locational choice for commercial investment, often being agricultural, and building construction in the cities [13,14]. Vacant lands, which are primarily transformed into housing on a daily basis, are increasing sprawling growth and property costs in the urban periphery and surrounding areas.

As a developing country in the world, Sri Lanka has been studied to define urban sprawl [4] and its impact [15] particularly in Colombo city and the spatiotemporal patterns of urban sprawl in Kandy city [8]. Besides these, the Batticaloa municipality area is highly accompanied by residential and commercial development. According to the Sri Lanka National Physical Planning Policy and Plan 2010–2030, Batticaloa is a rapidly developing city in Sri Lanka that is expected to become a metro city by 2030. Considering the existing situation in the Batticaloa municipality, many ongoing urban development projects have been carried out since the end of the Civil War in 2009. However, these developments were not well designed by planning experts. It led to less effective growth and changes in the urban area. Some studies have examined the urban land use changes in this area [16,17], and these studies provide an insight into why this area should be necessary for studying urban sprawl patterns. The built-up development has been growing rapidly in this area in recent decades [16], creating an irregular pattern or sprawling growth established with empirical evidence.

Furthermore, the characteristics of urban sprawl have been studied in various cities around the world (see Table 1), such as India, Malaysia, China, Romania, and the United States. However, these studies have not identified all the characteristics of urban sprawl in a single city. Several characteristics were only identified in a particular city in a developed country or a developing country. Moreover, since Sri Lanka is a developing country, it is difficult to identify influencing characteristics of urban sprawl based on the experience of previous literature from developing countries. Furthermore, cities in Sri Lanka never studied the characteristics of urban sprawl before, which also makes it difficult to understand the pattern of urban sprawl in the Batticaloa city. More precisely, there is still a lack of knowledge, especially about the characteristics (forms and patterns) of urban sprawl via the analysis of the different parts of the city, which is essential to tackle the sprawl effectively. The built-up patterns are the key parameter to identify the different characteristics such as low-density development, leapfrog development, commercial ribbon development and scattered development to establish the urban sprawl development. So far, the planners or academics have not yet addressed the sprawling development in this city in Sri Lanka. If this growth continues in this city, it will affect its sustainability when it becomes a metro city in 2030. In the end, this study can answer which characteristics influence the Batticaloa municipality through geospatial analysis.

Thus, this study can involve finding the different characteristics of urban sprawl in the Batticaloa municipality through spatial patterns. This finding can minimise the sprawling growth in the future and develop a sustainable city in Sri Lanka. The influencing characteristics in this city can be identified from the experiences of previous studies of urban sprawl in different cities in the world (see Table 1). Therefore, this study aims to identify the characteristics of urban sprawl patterns in the Batticaloa municipal council using Geographic Information System (GIS) and remote sensing technology. The findings can provide knowledge about the characteristics of urban sprawl to understand the sprawling patterns in other cities in Sri Lanka that have not been addressed so far. Apart from the empirical and methodological contributions, the findings of this study, in line with

Sustainable Development Goal 11 and the New Urban Agenda, offer useful insights and measures to control sprawl.

**Table 1.** Summary of characteristics for urban sprawl based on previous studies.

| Authors | Low Density | Leapfrog Development | Commercial Strip or Ribbon Development | Scattered Development | Auto Dependent or Car Dependent | Uncontrolled Growth | Uncoordinated Growth | Unplanned Growth |
|---|---|---|---|---|---|---|---|---|
| Hamad [1] | | | | | | X | X | X |
| Grigorescu et al. [5] | X | X | | | | | | X |
| Lv et al. [7] | X | | | X | X | | | |
| Yue et al. [9] | X | X | | | | | | |
| Paul et al. [12] | X | X | X | X | | | | |
| Farooq & Ahmad [13] | X | X | X | | | | | |
| Prakasa, Soemardiono, & Defiana [18] | X | X | X | X | | | | |
| Galster et al. [19] | X | X | | | | | | |
| Sudhira & Ramachandra [20] | | | | | | X | X | X |
| Ottensmann [21] | X | X | X | X | | | | |
| Shirkhanloo [22] | X | | | X | X | | | |
| Nikolov [23] | X | | | | | X | | X |
| Sinha [24] | X | | | | | X | X | X |
| Bhatta et al. [25] | | | | X | | | | |
| Osman, Nawawi, & Abdullah [26] | X | X | X | | | | | |
| Johnson [27] | X | X | | | | | | |
| Pichler-Milanović [28] | X | | | | | | | |

## 2. Definition and Characteristics of Urban Sprawl

Although urban sprawl is still considered an elusive concept, it has been used around the world for almost eighty years [29]. It was first realised through the transformation of agricultural and forestry areas into industrial, residential, and commercial development in the United States in the late 1950s. The term "urban sprawl" appeared in printed documents in 1960 [30] and is used in various fields, such as urban studies, remote sensing, and geography [18].

Urban sprawl has been defined by its characteristics identified in a particular urban area. Urban sprawl is the encroachment of non-urban lands to urban lands that occurred beyond the built-up area with a leapfrog pattern, unorganised pattern, low-density pattern, and unordered development [9]. Some patterns, such as ribbon development, low-density development, and leapfrog growth, were identified in Aligarh city, India, called urban sprawl [13]. However, urban sprawl was defined by eight metrics for land use, such as concentration, clustering, proximity, mixed-use, nuclearity, density, clustering, and centrality [19]. Sprawl is the irregular urban form with different scales of land use that consist of common and institutional facilities [31].

In contrast, similar patterns, such as uncoordinated, uncontrolled, and unplanned growth, were found along highways, called urban sprawl [20,32]. In addition, urban sprawl is unplanned discontinuous growth categorised by low-density growth in urban boundaries [5]. It is a typical pattern with scattered growth [11,21,22]. Urban sprawl is car-dependent, and low-density development has several negative impacts [22]. Low-density patterns and stripe development along major highways were identified in Colombo metropolitan area, Sri Lanka [4]. Sprawl development was identified in three directions along three main roads in Kandy city, Sri Lanka [8], which is almost the similar pattern of Colombo city.

Thus, the definition of urban sprawl varies among researchers who define it based on the characteristics of their urban area [5,7,19,22–24]. Although specialists and researchers still have problems defining the term "urban sprawl" [15,33]. However, many researchers accepted Ewing's (1997) definition is more suitable for recognising the urban sprawling, which stated an urban land use existence, characterised by scattered development, low-density, leapfrog development, and commercial strip development [12,34]. According to Table 1, more than 17 researchers have used more than one of these characteristics to explain urban sprawl.

Based on the review (see Table 1), urban sprawl characteristics, which are low density, leapfrog development, commercial strip or ribbon development, scattered and dispersed development, auto-dependent or car-dependent, uncontrolled growth, uncoordinated growth, and unplanned growth, were mainly identified in the developed and developing countries. Some characteristics, such as low density, leapfrog development, and scattered development (see Figure 1), were found in many countries [12,18,21], for example, Indonesia and India. As a developing country in the world, Sri Lanka has many development plans implemented due to the rapid urban growth in recent decades, which is causing urban sprawl.

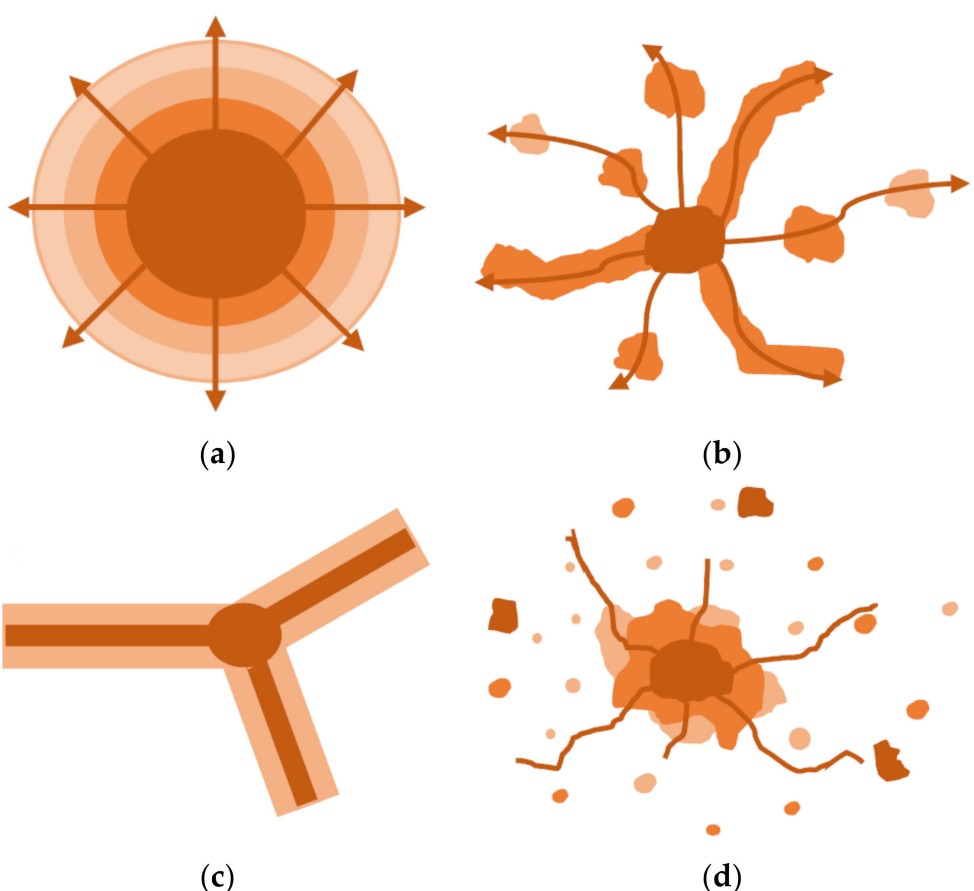

**(a)**

**(b)**

**(c)**

**(d)**

**Figure 1.** (**a**) Low-density development [35], (**b**) Leapfrog development [35], (**c**) Commercial ribbon development [35] and (**d**) Scattered development [36].

## 3. Materials and Methods

### 3.1. Description of Study Area

The study area, Batticaloa Municipal Council, is the local authority in the Batticaloa district located in the eastern part of Sri Lanka. The average elevation of Batticaloa is 8.523 m above Mean Sea Level. The total population in this area is 93,306 people. The

propagation of ethnic population is as a "sandwich pattern" with Tamil (91%), Muslim (5%), Sinhalese (0.14%), and others (3.86%) [37,38].

The total land area of the Batticaloa Municipality is 4311.87 hectares, which is allocated for different purposes of utilisation such as residential, agricultural lands, commercial, wetlands, water bodies, scrub forests, and others (refer to Table 2). Five (5) land parcels separate this city from the inland water bodies. These land parcels are connected to the bridges for transportation. As a clustered city, the land area uses for multiple purposes and these links with various sectors, such as fishing, agriculture, small industries, and commercial. Each land parcel has different property uses and has cluster development in each sector, such as commercial, recreational, and residential. The most dominant land use of this area is residential (1170.24 hectares), and the next is agricultural lands (935.6 hectares). One of the low proportions of the land-use class is commercial (23.5 hectares), which compares to other major land uses. The natural lands, including wetlands (82.5 hectares), water bodies (58 hectares), and scrub forests (185.71 hectares), are also a certain portion in this area [37].

**Table 2.** Major land use categories in Batticaloa Municipality.

| Land Use Type | Area (Hectares) | Area (%) |
| --- | --- | --- |
| Residential | 1170.24 | 27.1 |
| Agricultural lands | 935.6 | 21.7 |
| Commercial | 23.5 | 0.6 |
| Wetlands | 82.5 | 1.9 |
| Water bodies | 58 | 1.4 |
| scrub forests | 185.71 | 4.3 |

Considering the infrastructure facilities in the Batticaloa municipal area, it is undeniable that ongoing development projects will improve their current conditions. However, these development activities are not planned by urban planners and relevant development officials. Arbitrary development occurs highly in this area which affects the pattern of sustainable land use. Since 1990, the rapid growth of the population has caused several changes in the built pattern that were not assessed by the authorities for sustainable development. Permanent and temporary migration to Batticaloa municipality increased from other parts of the district and the Eastern province due to the effects of the Civil War because this area is the major urban centre in the Eastern Province, Sri Lanka, with all amenities. In addition, the proposed development plan in Batticaloa municipality for 2030 contained many rules and regulations on land use, especially built-up development, which is not considered much further in current development. Thus, Batticaloa municipality began to face the urban sprawl development in the core city and the periphery. As rapidly developing cities in Sri Lanka, Batticaloa received more concern from planners as the city was expected to become a metro city by 2030. Therefore, this area has been chosen to study the urban sprawl in Sri Lanka, which is more significant to understand the characteristics of sprawling. Figure 2 shows the Batticaloa Municipal Council area, the study area in this research.

### 3.2. Data Collection—Source of Data

Understanding the characteristics of urban sprawl requires the pattern of land use, especially the built-up changes in the area. Google Earth Pro and ArcGIS 10.6.1 applications were utilised to produce various layers, such as built-up maps, density maps using satellite images, boundary map for Batticaloa municipal council, and Grama Niladhari division map for Batticaloa Municipality.

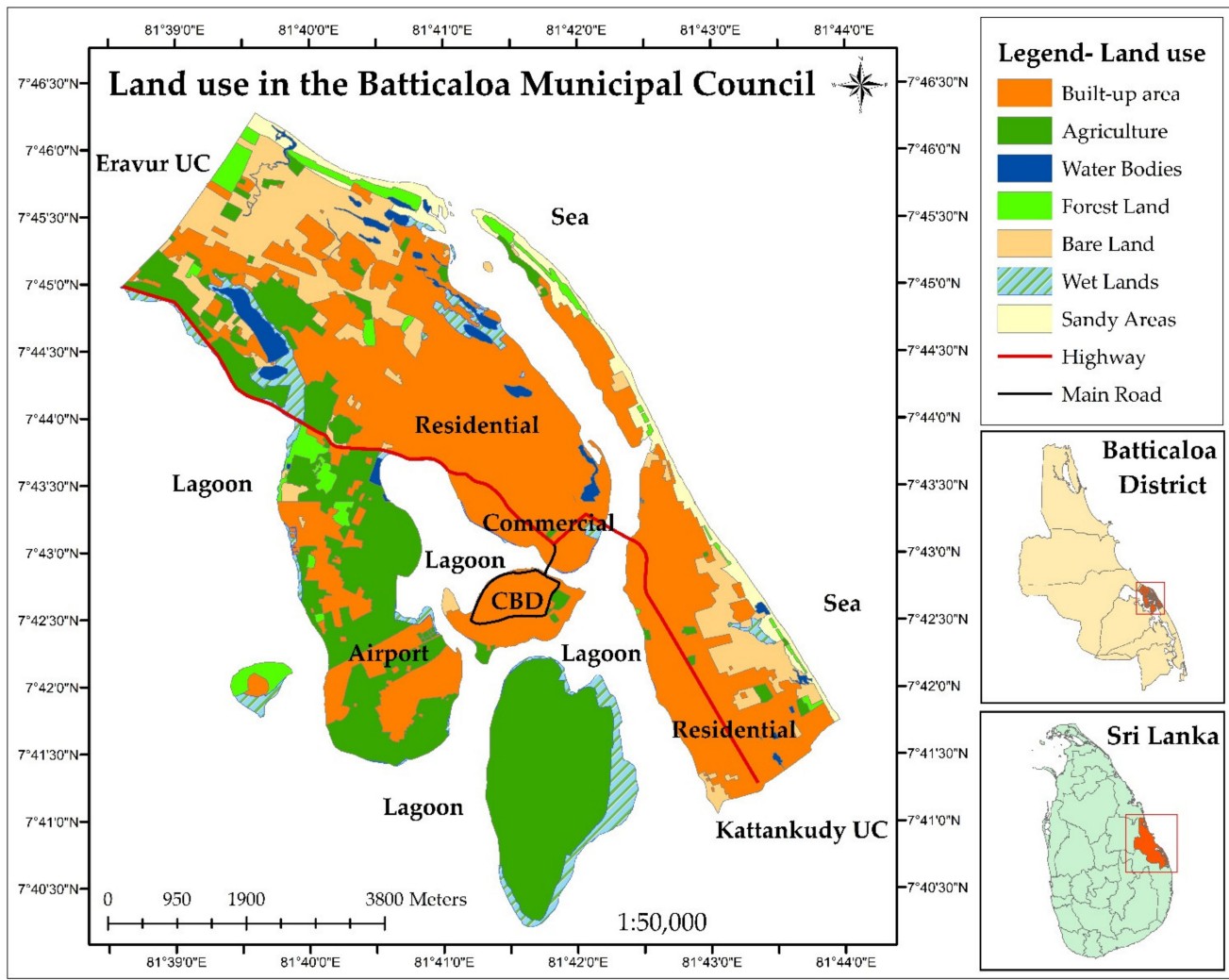

**Figure 2.** The Study Area—Land use pattern in the Batticaloa Municipal Council—2020. Source: Modified from the Batticaloa Municipal Council Profile, 2021.

There are two (2) types of data, namely, remote sensing data (satellite images) and demographic data, used to generate the maps. Satellite images were downloaded from the Earth Explorer, United States Geological Survey [39]. Details of this data presented in Table 3 with all the information. These time-series images for 2000, 2010, and 2020 used to produce the land use maps to identify the sprawling and extract the built-up area. Meanwhile, the demographic data were collected from the Department of Census and Statistics, Sri Lanka. This data was utilised to produce the population density map in order to identify the density changes. Then, census data for the years 2001, 2012, and 2019, which are the most recent years with satellite images, were compared with the built-up images to understand the influence of population growth on urban sprawl.

**Table 3.** Details of satellite imagery data.

| Type of Satellite | Image ID | Acquisition Date | Resolution |
|---|---|---|---|
| **Landsat ETM+ (2000)** | LE07_L1TP_140055_20000928_20170209_01_T1 | 28-SEP-00 | 30 m |
| **Landsat TM (2010**) | LT05_L1TP_140055_20100924_20161212_01_T1 | 24-SEP-10 | 30 m |
| **Landsat 8 (2020)** | LC08_L1TP_140055_20200303_20200314_01_T1 | 03-MAR-20 | 30 m, Pan-15 m |

Source: Earth Explorer, 2021.

### 3.3. Data Processing and Analysis

The downloaded satellite imageries were geo-referenced in World Geodetic System 84 (WGS84) and then projected to the Kandawala local coordinate system. Filters, brightness, and contrast tools were used to improve the quality of the satellite images. The Batticaloa Municipality's boundary was digitised as a shapefile using the current map to demarcate the study area. Based on this boundary, three (3) satellite images, which are from the years 2000, 2010, and 2020, were clipped to delineate the satellite images based on Batticaloa Municipality's boundary. Then, these images were classified into six (6) classes, namely, built-up, agriculture, forest, water bodies, vacant land, and others, according to the training land samples using Supervised Maximum Likelihood classification in ArcGIS. Then, the classified images were validated using the accuracy assessment technique. About 85% of overall accuracy is usually considered enough in the map data [25]. The overall accuracy of land use can be obtained by the following Equation (1) [40]:

$$OA = (1/N) \sum_{i=1}^{r} n_{ii} \tag{1}$$

where $OA$ is overall accuracy, $n_{ii}$ is correctly classified pixels' number, N is pixels' total number, and r is rows' number.

Ground truth data was considered in comparing the classified Landsat images. Overall, 127 training samples for ground truth were obtained as random points at specific locations using the grid layout in Google Earth Pro and using known coordinate points. Each point has valid land-use class values, which are:

(1) Built-up;
(2) Agriculture;
(3) Water bodies;
(4) Scrubland;
(5) Mangroves;
(6) Vacant land;
(7) Others, including the playground, transportation, park, and public land, for the classified and ground truth fields.

The confusion matrix was formulated to find the accuracy and obtain individual accuracy between the classified classes and the reference data, such as coordinate points collected from the field. The classification accuracy was determined to obtain the level of precision. The final output of this process was a land use map of Batticaloa Municipality from the years 2000, 2010, and 2020.

Then, density mapping was used, which is a method to show the location of points or lines which can be concentrated in a given area. Such maps often use interpolation methods to estimate a given surface where the concentration of a given function can be. The population density is 1372 persons per km$^2$ in Batticaloa [37], which is rapidly increasing in recent decades due to rapid urbanisation. Therefore, the population density map was produced using the density analysis tool in ArcGIS. The population for each Grama Niladhari divisions was utilised to produce the density map. The spatial boundary data for Grama Niladhari was developed using the base map of the Batticaloa Municipality. The population data for each Grama Niladhari division were added to the spatial file to show the population's spatial distribution. Based on the standard calculation of the population density, the number of people divided by land area is calculated. The Equation (2) for population density is as follows:

$$PD = (Tn/a) \tag{2}$$

where $PD$ is population density, $Tn$ is the total population in a particular area, and $a$ is the land area in hectares.

It is categorised by suitable data clustering methods, which is Jenks's natural breaks classification, designed to determine the best arrangement of values. It is used to classify

the data into five (5) categories, such as very high, high, moderate, low, and very low. This density distribution helps to understand the changes clearly as to which area is high and low density. Figure 3 shows the complete methodological framework for identifying urban sprawl characteristics.

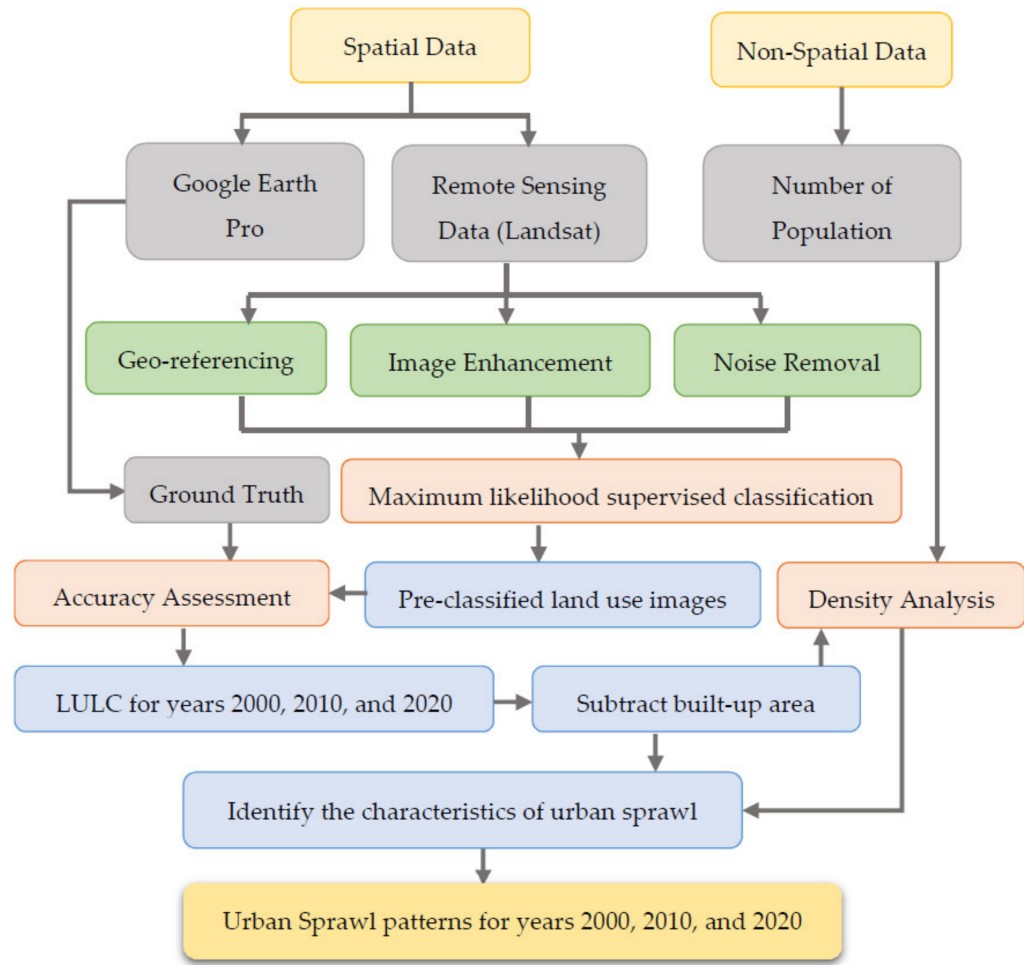

**Figure 3.** The methodological framework for identifying urban sprawl characteristics.

Further, the built-up density map was also produced to establish the low-density development. The buildings of the study area were digitised using Google Earth Pro. Each building's features were converted into points using the feature to point tool in ArcGIS. Then, buildings for each Grama Niladhari divisions were clipped, and density analysis for buildings was conducted to generate the built-up density for the Grama Niladhari division in Batticaloa municipality. The density was categorised by Jenks's natural breaks method into five (5) classes as very high, high, moderate, low, and very low. The density changes in the study area were identified by using these maps. Based on the results also, we can clearly understand which area has more sprawl.

## 4. Results and Discussion

The dynamics of the built-up area, known as a typical process of urban sprawl, are of particular importance for understanding spatial patterns for sprawl development. The built-up spatial patterns identified the characteristics of the urban sprawl in the Batticaloa Municipal Council.

### 4.1. Built-up Patterns

The built-up patterns are presented in the maps of years 2000, 2010, and 2020 (refer to Figure 4) to understand the sprawling characteristics in the Batticaloa municipal council. The built-up area has an extent of 1162 hectares in 2000, which increased to 1439 hectares in 2010. It increased to around 1557 hectares in 2020, showing the increases in the city's built-up pattern (refer to Table 4). Based on this, a rapid increase in built-up growth was identified during the selected periods.

**Table 4.** The extent of built-up area in Batticaloa Municipality.

| Category | 2000 | 2010 | 2020 |
|---|---|---|---|
| Built-up area | 1162 Hectares | 1439 Hectares | 1557 Hectares |

The accuracy of the built-up pattern is identified by the classified land use maps of the study area. The analysis revealed that the producer accuracy and the user accuracy varied during the selected periods. Producer accuracy refers to the accuracy of the map with how often real features on the ground are displayed correctly. User accuracy refers to the map user accuracy, which indicates how often the land use class on the map is actually present on the ground. User accuracy shows the reliability of the map. According to the calculation, the producer accuracy for the built-up area in 2000 is 89.43%, while the user accuracy is 96.72%; in 2010 the producer accuracy is 94.6%, while the user accuracy is 99.72%; and in 2020, the producer accuracy is 87.71%, while user accuracy is 93.29%, which is an excellent precision for analysis. The overall accuracy of around 85% is considered enough to prove the precision of the map data [25].

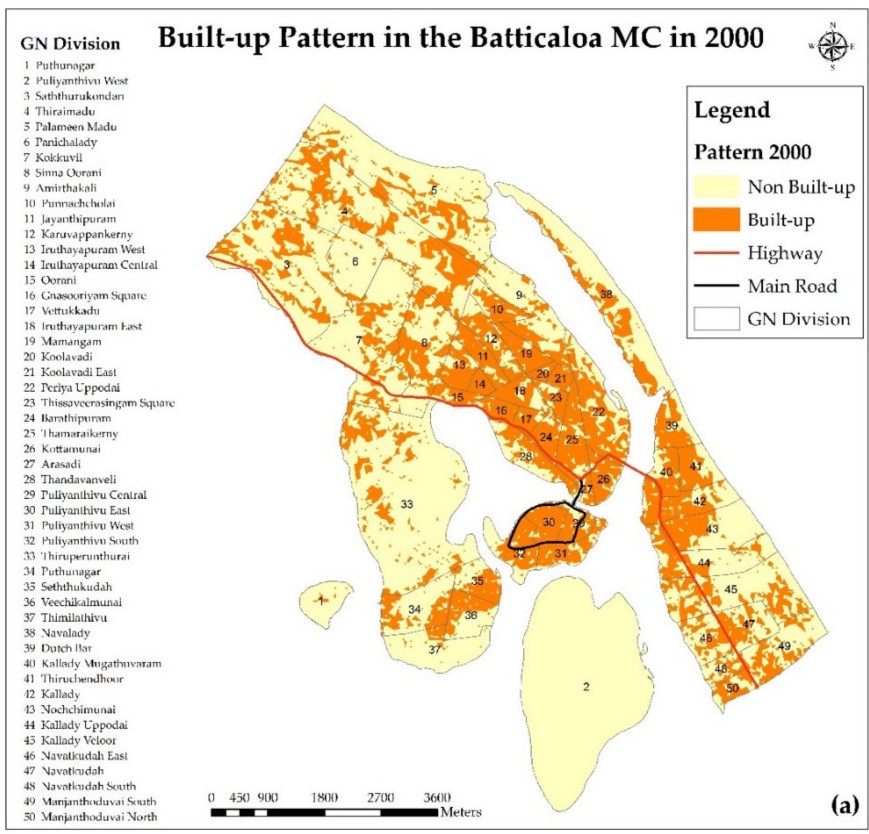

**Figure 4.** *Cont.*

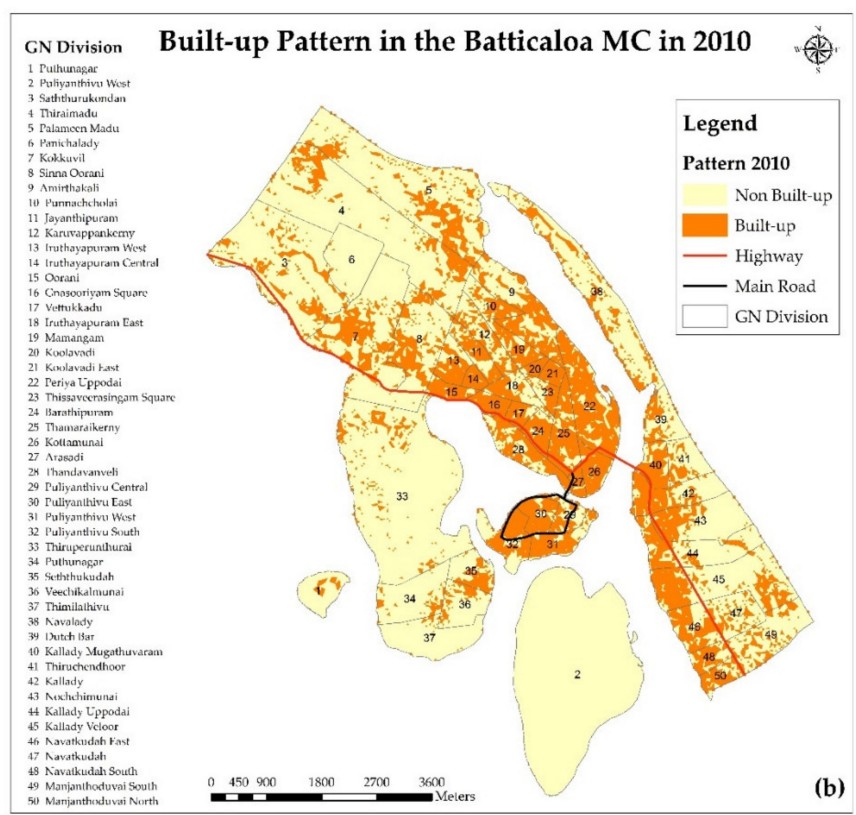

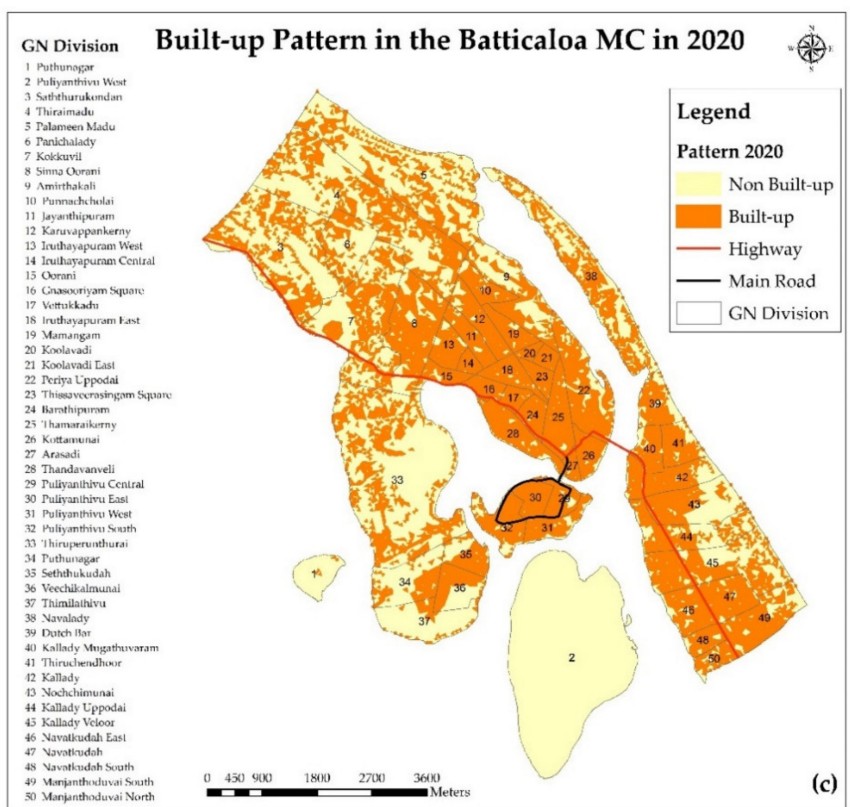

**Figure 4.** Built-up patterns in the Batticaloa Municipal Council in (**a**) 2000, (**b**) 2010, and (**c**) 2020. Note: The Grama Niladhari Division (GN) is a subdivision of the Divisional Secretariat in Sri Lanka. A total of 14,022 Grama Niladhari divisions are in charge of 331 Divisional Secretariat divisions in Sri Lanka; of these, 48 of Grama Niladhari divisions are in the Batticaloa municipality.

The Batticaloa municipality area consists of 48 Grama Niladhari divisions. Thiraimadu is one of the divisions that emerged with housing developments after the tsunami disaster. This area is also developing as an administrative zone in the city that encourages people to construct housing, which is one reason for rapid built-up development in this area. In addition, Puliyanthivu Island is the city centre, with densely developed commercial and residential buildings. However, the lower land value in the Navalady and Thiraimadu areas attracts people with low and middle incomes to buy the land and build a house. The main reason for the lower land value is that this area is often affected by disasters, especially flood. Besides this, high land value has been identified in the Puliyanthivu, Oorani, and Thandavanveli areas because these areas are close to the city centre, the highway, and several infrastructure facilities. One of the best policies is Land Value Capture and Taxation, which is beneficial for affordable housing in this city with a lower land value. This system is in place to increase revenue and fix up the downtown buildings in the city. This income can be used to develop housing for people with low incomes. However, the tax system is already implemented in the city, which is not strictly followed annually. Although everyone is aware of the property tax in the city, they sometimes forget to pay the annual renewal tax. The municipality does not remember and observe these activities regularly, which leads to the illegal land formation as well as sprawling development in the city.

Several groups of people own the total land area in the Batticaloa municipality. This land has been distributed around 73.1% to the inhabitants, 9.1% to the government, 4.7% to Batticaloa municipality, and 13.1% of obscure details. One of the principal regulations is that people cannot construct any buildings in the Batticaloa municipality area without obtaining an approved development permit. However, some people carry out illegal construction development, which represents around 13.1% of the total area of Batticaloa municipality, and those do not contain explicit information about the property. These owners build houses or other buildings on other people's land without getting the proper approval from the municipality. These activities increase the most illegal construction in the city. These developments triggered the formation of scattered and leapfrog development in the city, which are the main reasons for the sprawling growth in this city. Therefore, the property documentation system should be appropriately maintained by the municipality. A survey for property owners should be conducted at the turn of the year to identify illegal land and minimise sprawling growth. This survey can also inspire people to pay taxes without fail. In addition, the municipality should gently remind people before the annual tax period ends. For this, a smart application should be developed for the municipality to identify the pending cases quickly. These practices can control land occupancy and the maintenance of more than one piece of land of a person in the city.

In addition, land ownership problems related to communities in the Batticaloa municipality were identified in the Nochchimunai and Sinna Oorani Grama Niladhari divisions. At the same time, the Puliyanthivu South, Kallady uppodai, Kokkuvil, Punnaicholai, Karuvappankeny Amirthakali, and Mamangam areas identified landowners' issues associated with low-income people. These types of problems caused to form vacant land and more subdivided lands in the city. In addition, illegally divided lands were identified mainly for sale in the Saththurukondan and Uppukarachai area, where the land value is relatively high today. These activities mainly lead to sprawling growth in the city. Therefore, the municipality must circulate a pre-approval method to divide the land in the city. People should inform the municipality about the subdivision of the land, and then the municipal official should visit the specific area to observe the land. After that, the municipal guidelines must be adapted to that particular area, and the entire previous land document must be verified to confirm the land entitlement for subdivision and sale. This practice can find illegal subdividing of land and sale, minimise future land problems within communities, and reduce the amount of vacant land in the city.

Figure 5 shows the built-up changes between the years 2000 and 2010, and between 2010 and 2020 to understand the expansion. This comparison showed a gradual increase in built-up changes in the study area. The built-up growth increased 227 hectares between

2000 and 2010, and 118 hectares between 2010 and 2020 (see Table 5). These gradual changes in the built-up area have illustrated the growth of sprawling in the city. The rapid growth was registered between the years 2000 and 2010. The Batticaloa area was one of the severely affected areas by the civil war in Sri Lanka. Thus, most people from the other parts (rural areas) of the Batticaloa district, such as Porativu, Vellaveli, Mandur, Thikkodai, and Vaharai, migrated to the Batticaloa city for survival, including security, education, and livelihood. Furthermore, the living standards, access to more facilities, and admired city life are the reasons for the migration of people to the city. The movement to the city led to the demand for land and housing for the people. The desire for their own property has created a haphazard development in the city and the urban fringe of the Batticaloa area. Thus, different characteristics were identified in the different areas of Batticaloa city. However, many urban sprawl characteristics exist in the world's cities, but the study area is occupied with some characteristics.

**Table 5.** The changes of built-up area in Batticaloa Municipality.

| Category | Timespan | Expansion |
|---|---|---|
| Built-up Area | 2000–2010 | 277 Hectares |
|  | 2010–2020 | 118 Hectares |

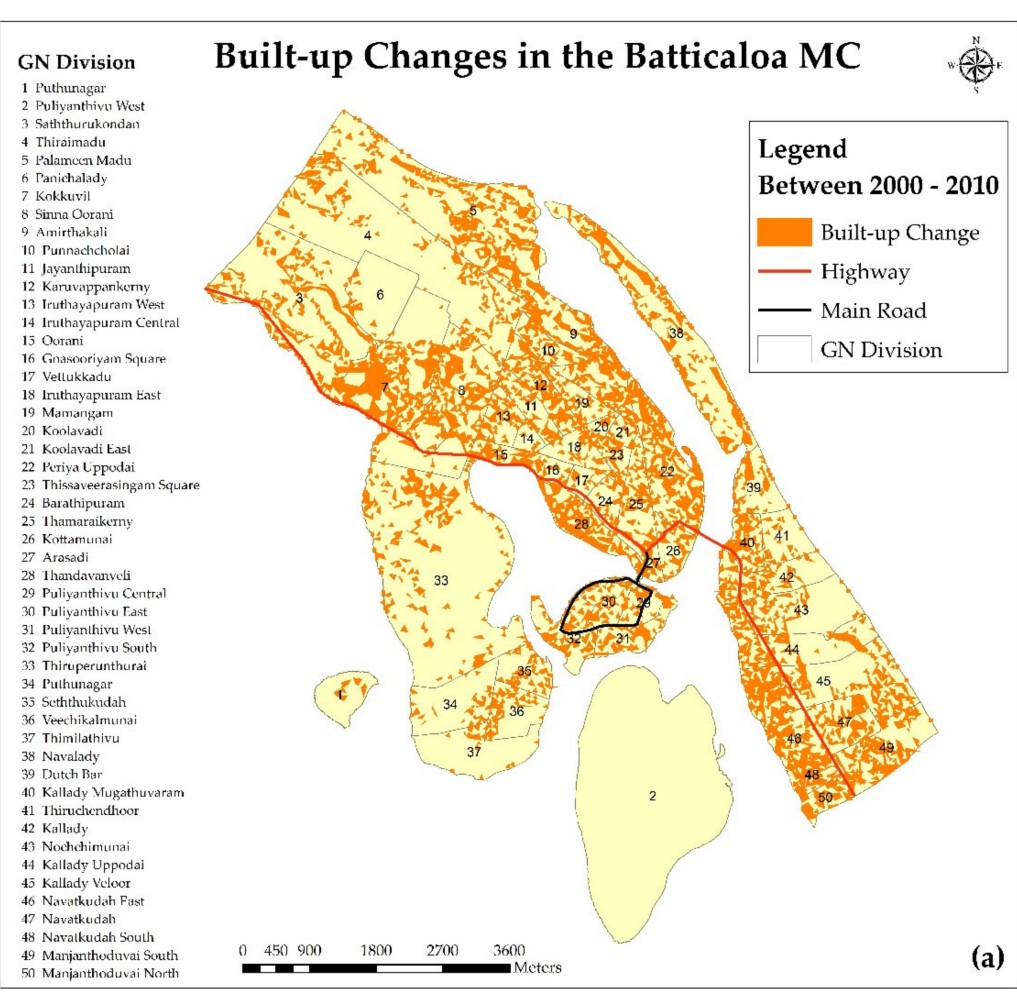

**Figure 5.** *Cont*.

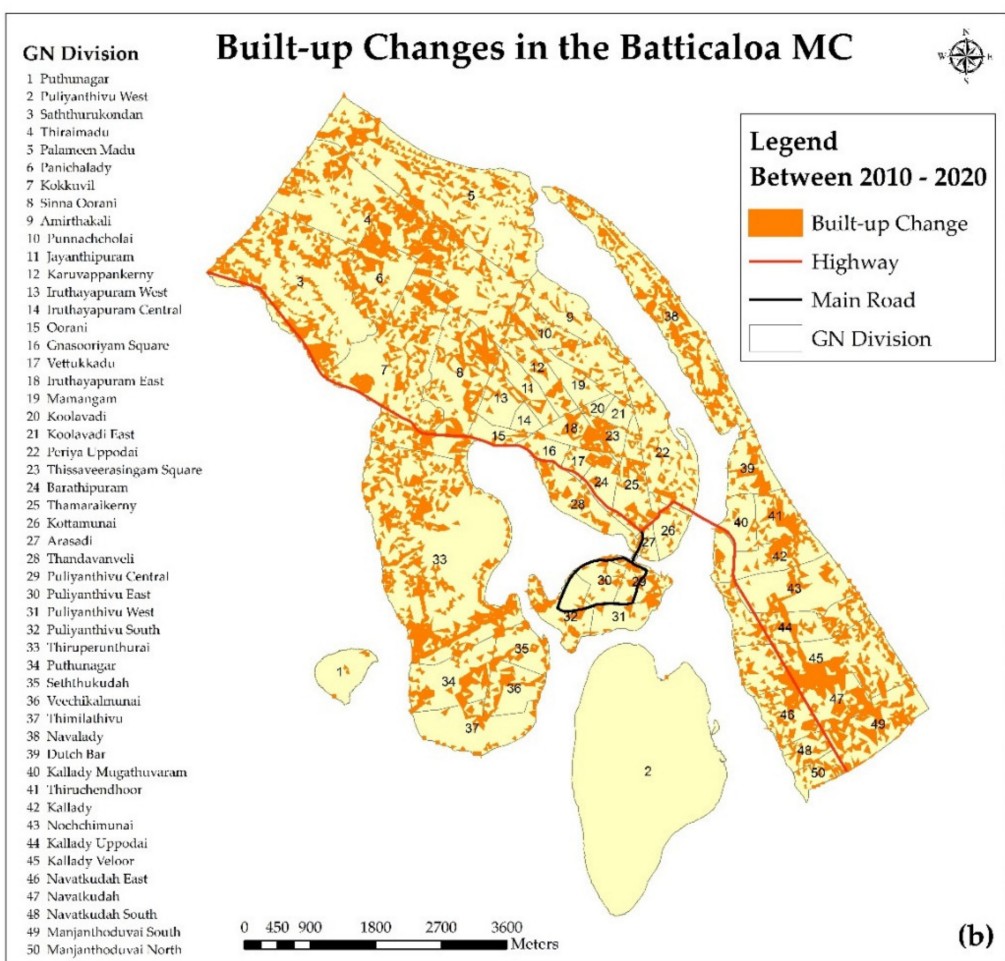

**Figure 5.** Built-up changes in the Batticaloa Municipal Council (**a**) between 2000 and 2010, and (**b**) between 2010 and 2020.

### 4.2. Characteristics of Urban Sprawl

The built-up area was extracted from the land use map to identify the urban sprawling characteristics. The spatial and temporal built-up patterns reveal that sprawling characteristics identified as low density, leapfrog development, scattered growth, and commercial ribbon development influenced the irregular urban development pattern. Most of these characteristics are identified in the city limits, and some are in the core city, which affects the city's sustainable growth.

#### 4.2.1. Low-Density Development

Low-density development is one of the main phenomena of urban sprawl generally risky to the urban environment. The primary units for identifying the urban sprawl, including density, are buildings, especially housing units of a particular area [19]. The residential developments are mainly identified in the marginal low-density areas in the city. Low-density development is a piecemeal extension of the built-up area, which consumes much land in the urban fringe. It is the most generally indicated characteristic of urban sprawl in many pieces of literature [41]. Residential housing mostly consumes the vast land, which was vacant land previously, leading to the low density. The rise of land and property value in the city cannot afford a vast population; however, this value is meagre in the urban fringe. Thus, the sprawl areas are occupied mostly by the low-income people for their permanent residence. They are attracted by these vast, spacious living areas to build an affordable house [24], also experienced by the Batticaloa area. The housing preference of the lower class and some middle-class people pushed them to settle in these low land

value areas. The people who migrate from the village areas admired the city limits, which is more spacious and affordable for their own housing. In addition, a single dwelling unit in a larger area in the Batticaloa municipality is one of the main reasons for the low-density development, which is similarly identified in the United States of America [24].

Figure 6 shows the built-up density in the Batticaloa municipal council area by Grama Niladhari division. Based on this, higher density patterns were identified in the city centre from 2000 to 2020. However, nine divisions, which are Saththurukondan, Thiraimadu, Paalameenmadu, Panichalady, Kokkuvil, Thiruperunthurai, Thimilathivu, Veechikalmunai, and Navalady, mainly come under the low density in the divisions throughout these periods. The lack of space in the city centre for housing development has been limited to low-income residents where land value is most in demand. Further, nearly 150 low-income families living in the city had the rural characteristics identified in Sinna Oorani, Punnaicholai, Thiraimadu, Mamangam, Kokkuvil, and Saththurukondan areas. The developed built-in density map can be useful for identifying areas with low and very low density in the city. Based on this, an appropriate development plan can be adapted to this area to make the city more compact.

Table 6 shows the range of built-up density in the Batticaloa Municipality. The ranges were categorised from very low density to very high density. Based on this, around 0–4 buildings per hectare were identified in the very low-density areas and more than 22 buildings per hectare in the very high-density areas. The built-up density increased near the city centres, except in the northern and western part of the city, from 2000 to 2020. The main reason for the low density in these areas is the inadequate facilities such as accessibility to highways, commercial, and other services.

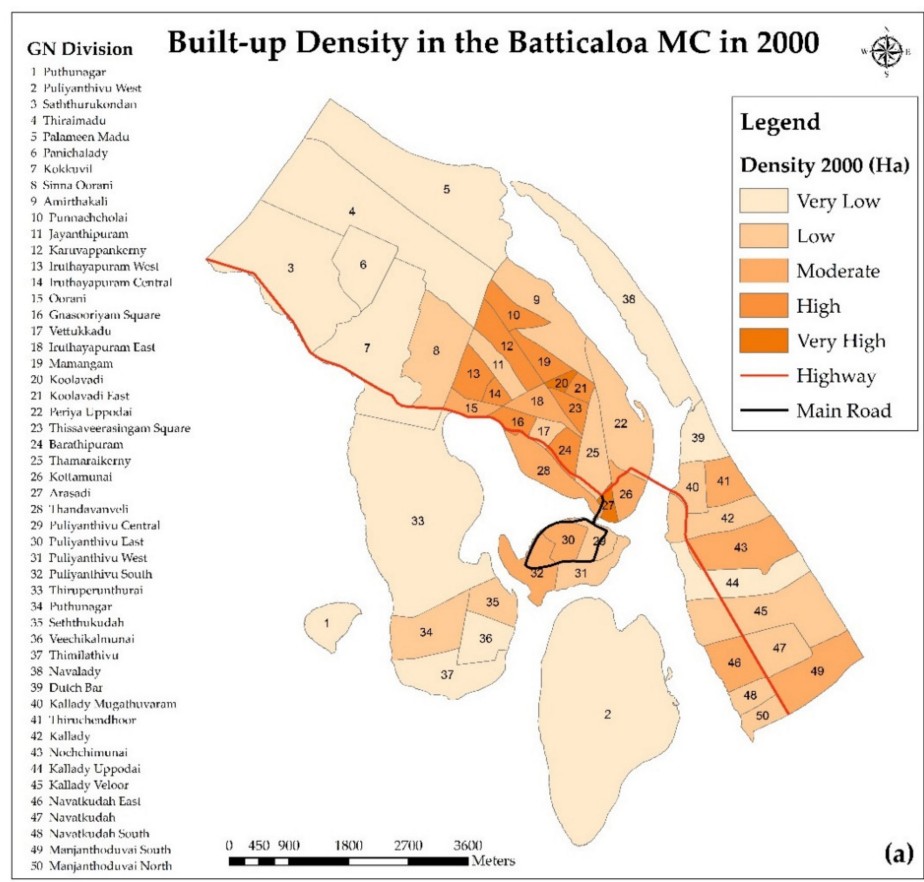

**Figure 6.** *Cont.*

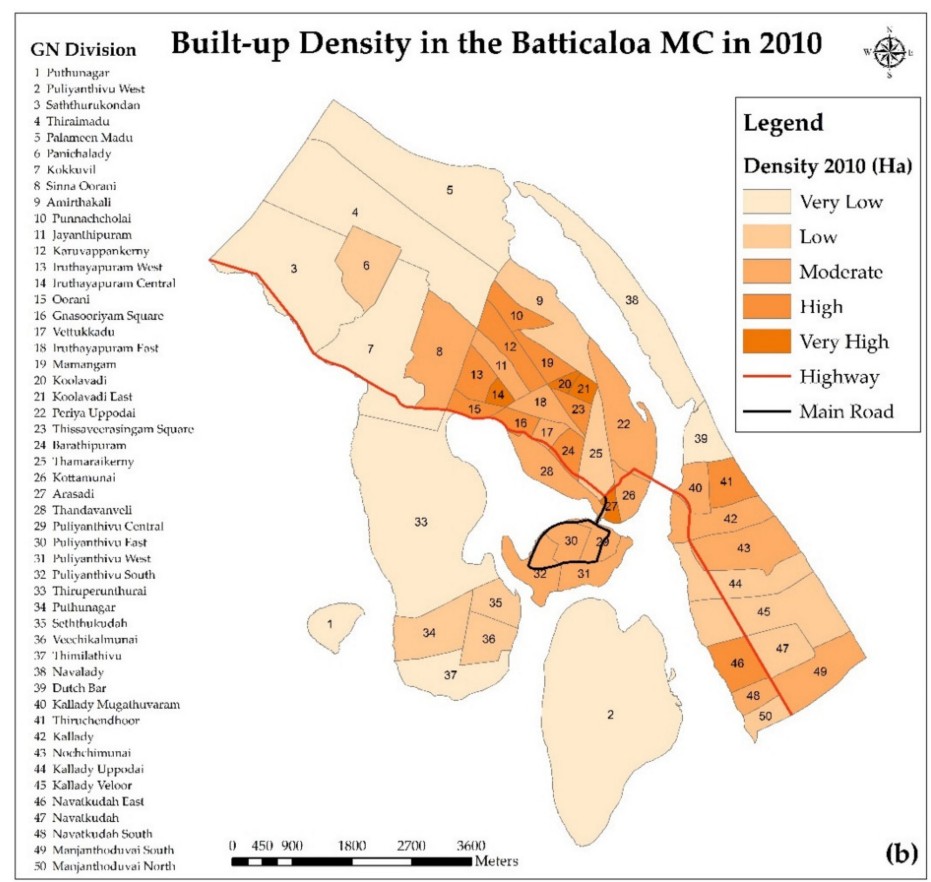

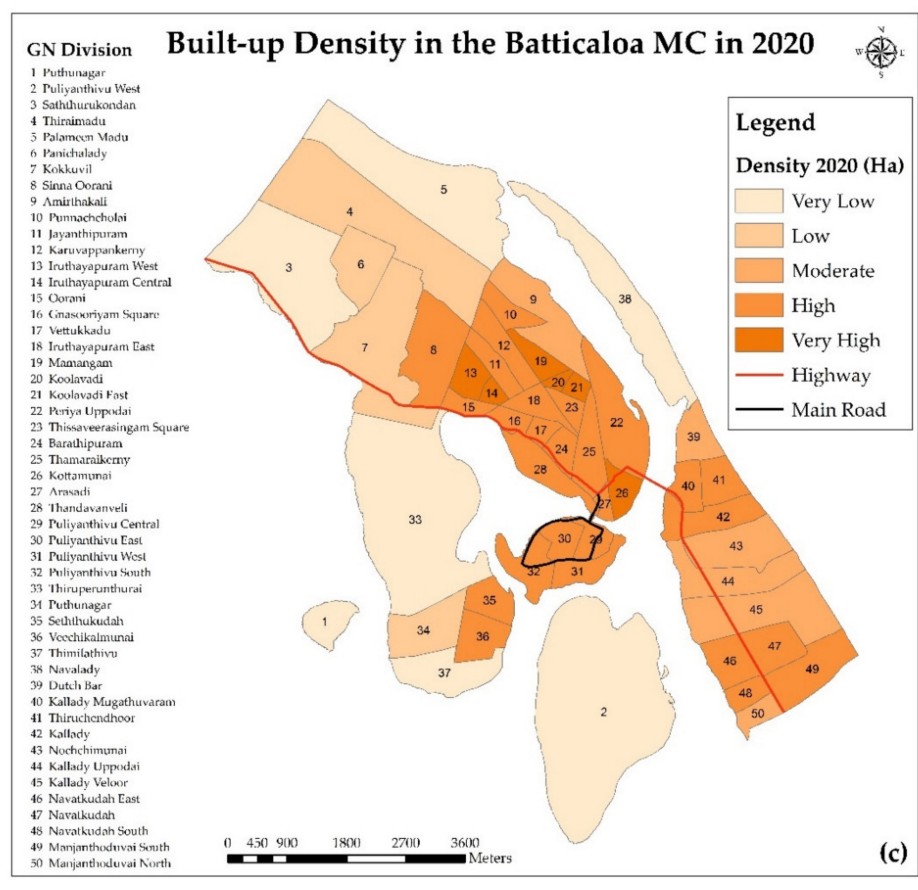

**Figure 6.** Built-up Density in the Batticaloa Municipal Council in (**a**) 2000, (**b**) 2010, and (**c**) 2020.

**Table 6.** The range of built-up density in Batticaloa Municipality.

| Scale | Density (Buildings/Ha) | | |
|---|---|---|---|
| | 2000 | 2010 | 2020 |
| Very Low | 0.0–4.0 | 0.0–4.0 | 0.0–4.0 |
| Low | 4.1–8.0 | 4.1–8.0 | 4.1–8.0 |
| Moderate | 8.1–13.0 | 8.1–13.0 | 8.1–13.0 |
| High | 13.1–22.0 | 13.1–22.0 | 13.1–22.0 |
| Very High | 22.1–37.0 | 22.1–31.0 | 22.1–34.0 |

Most of the lands are used for single use, like individual housing, which created the low-density development in the Batticaloa municipality. For example, 88% of homes are single-storey separated houses, 9% of homes are two-storey separated houses, and 1% of homes are more than two-storey separated houses. These housing patterns show a rural characteristic in this city. In addition, these single housing developments are one of the main reasons for the low-density development in this area. Therefore, housing policy must be designed in accordance with Sustainable Development Goal 11 and the existing situation of Batticaloa municipality. Housing policy should be developed in consultation with stakeholders in Batticaloa Municipality who provide a clear view of all income earners and the different communities living in the city. The municipality then displays the decision for public responses that provide different ideas for improving the policy before it is implemented. Furthermore, reporting the municipality's policy in public can also make the right decisions in all activities by people, including building houses and maintaining the land.

In addition, building codes must be provided to track building types and the location of buildings in the city. This method can help to quickly identify a specific building in all situations and demolish illegal constructions. These activities primarily help control the future sprawling growth of the city. Further, the developed built-up pattern and density maps are helpful to identify the additional unregistered buildings in each Grama Niladhari division. For example, a Grama Niladhari division already has 25 buildings registered in the municipality, but the map shows 28 buildings in the same division. By this, the constructions can be understood as illegal development in the area in question. The municipality can take the necessary measures against them and also minimise the sprawling growth in the city. In addition, a monitoring unit should be set up to review housing policy and the necessary strategies for building construction in the city. An online platform should be developed to guide public discussions and consultations. This continuous monitoring activity can control illegal housing development in the city.

The population density is measured by the ratio of people inhabiting a specific region in persons per square kilometre or hectare. A city that occupies a smaller land area is considered more compact and less sprawled, and that with more extensive land occupied by less population implies low density and a more sprawled characteristic [41]. Table 7 shows the range of population density in the Batticaloa Municipality. Population density ranges from very low to very high density. Areas with 0 to 10 persons per hectare are known as very low-density areas. Areas with over 100 people per hectare were identified as very high-density areas in 2001, and with more than 84 people per hectare were very high-density areas in 2012 and 2019. The population density is almost high in the city centre and close to commercial areas.

**Table 7.** The range of population density in Batticaloa Municipality.

| Scale | Density (Persons/Ha) | | |
|---|---|---|---|
| | 2001 | 2012 | 2019 |
| Very Low | 0.00–10.00 | 0.00–10.00 | 0.00–10.00 |
| Low | 10.01–34.00 | 10.01–29.00 | 10.01–29.00 |
| Moderate | 34.01–60.00 | 29.01–53.00 | 29.01–53.00 |
| High | 60.01–100.00 | 53.01–84.00 | 53.01–84.00 |
| Very High | 100.01–151.00 | 84.01–124.00 | 84.01–131.00 |

The consumption of the land is faster than the population growth, which revealed low-density development. As shown in Figure 7, the population density in the Batticaloa Municipality is identified by the Grama Niladhari divisions. Based on this, developed areas such as Puliyanthivu, Arasady, Thandavanveli, Oorani, Kallay, Iruthayapuram, Mamangam, Thiruchendhoor, and Navakkudah are almost high built-up density areas. However, those areas have not populated much when compared with built-up. However, the highest population density was identified in the Arasady, Koolavady and Iruthayapuram areas. Low density was registered in the edge areas, but the core city and the highway area only showed higher population density and built-up. Paalameenmadu, Navalady, Saththurukondan, Puliyanthivu west, and Thiruperunthurai areas were identified with the lowest population density patterns in the Batticaloa municipality. The main reason for the low density is poor accessibility to highway and other services. In addition, Seththukudah, Vechukalmunai, and Thimilaithivu areas were under military control during the civil war period. Thus, people did not desire to make their settlement in those areas.

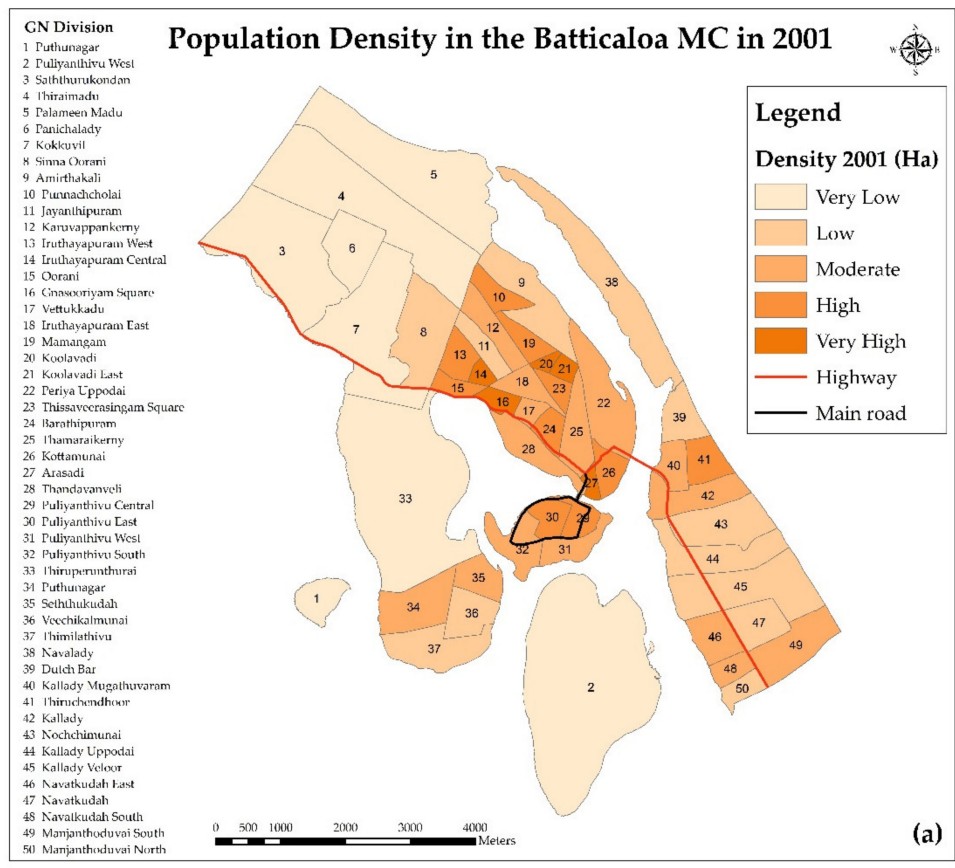

**Figure 7.** *Cont.*

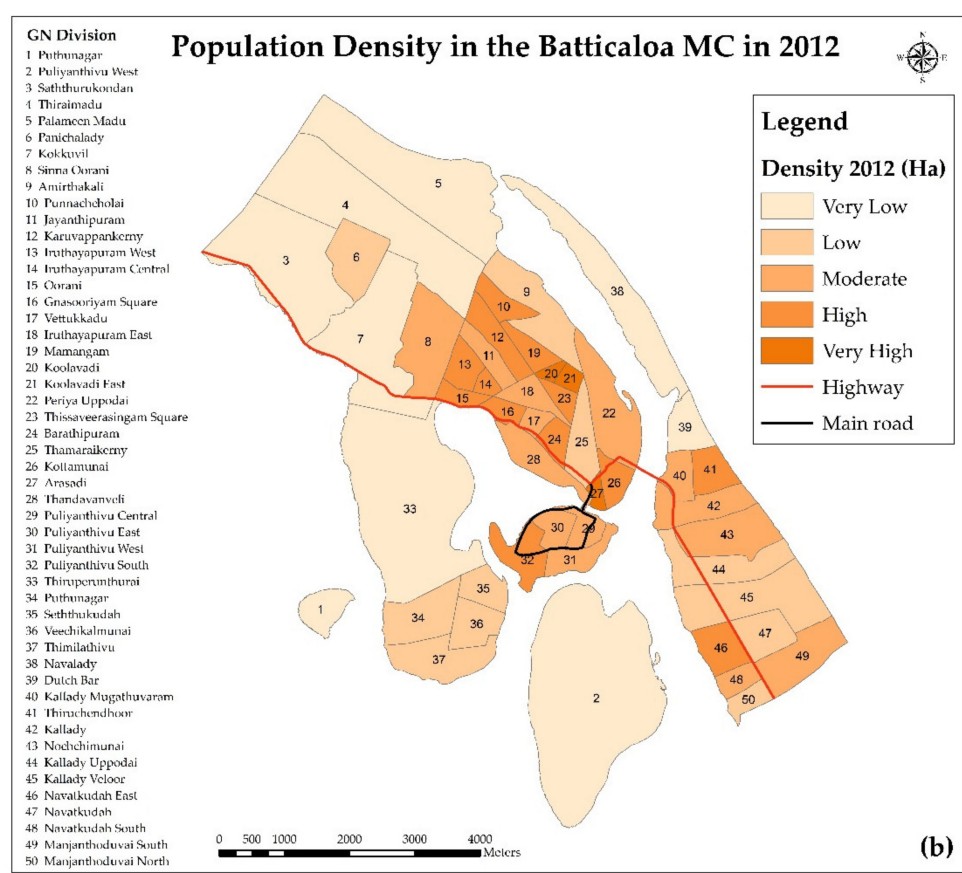

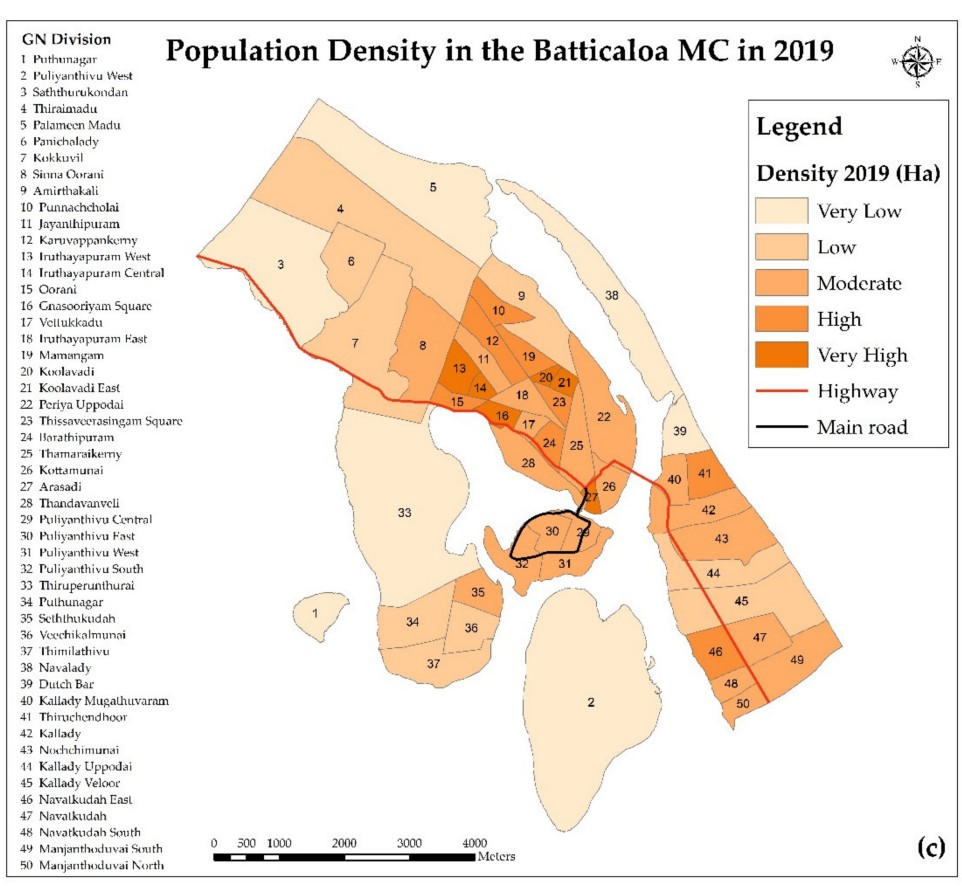

**Figure 7.** Population Density by Grama Niladhari Divisions in (**a**) 2001, (**b**) 2012, and (**c**) 2019.

Additionally, Batticaloa Domestic Airport is located in the Thirupperunthurai division, which is closest to the Puthunagar and Sethukkudah areas. Residential buildings are banned around the airport areas such as Puthunagar, Thirupperunthurai, and Sethukudah due to the airport expansion project. In addition, a water supply system was launched in the high-density populated areas such as the city centre and the nearby areas. In other areas, people use well water for their needs. Further, 38 schools are located in the Batticaloa municipality area. Of these, seven schools are national schools with a high level of education, located in the city centre and the nearest areas, which is one of the reasons for the high population density.

Further, the population of this city has a high growth rate of 3.92% in the period 1990–2001 and around 2.07% in the period 2001–2010. This rate changes to around 2.27% in the period 2010–2019, which indicates the fastest growth of cities in Sri Lanka. In addition, the population growth rate in the future is expected to be 2.5% to 3.0% in 2030. The minimum expected population is 127,291 persons and a maximum of 170,714 people in 2030 [37]. This rapid population growth can lead to more sprawling development when political influences disrupt municipality development plans and regulations. Therefore, the rules and regulations must be strictly followed to become a sustainable city in the future.

### 4.2.2. Leapfrog Development

A discontinuous irregular pattern on developed land is widely recognised within the city limits. This type of development makes it costly to provide essential services like water and drainage. This development consumed a wide range of land and created an arbitrary development pattern that destroys urban beauty. Figure 8 shows the leapfrog development, which creates more changes in the land use pattern, leading to the urban sprawl established in the visual map. This development can identify a very inefficient land use pattern, which is one of the most extreme examples of urban sprawl. Such growth affects the development of the city directly, including infrastructure and services.

The leapfrog development creates less housing and population density due to the undeveloped land, such as the urban fringe. This density is higher than the individual homes, which affects sustainable development [21]. The vacant land in specific areas such as Thiraimadu, Paalameenmadu, Kokkuvil, Panichalady, and Navalady are good examples of leapfrog growth with low density in the Batticaloa municipality area. Fundamental accessibilities such as public transportation, telecommunication, and water supply are comparatively poor, leading to less population growth. These people live in small housing units, which means one-room or two-room houses built using bricks or clay, but the land extent is larger than the houses. The people living in these areas are from low-income classes. The municipality should introduce more housing schemes, incorporating with the National Housing Development Authority, Sri Lanka. This development can reduce the leapfrog development within the city limits. In addition, the municipality should raise people's awareness about the leapfrog development and its impacts on the city through community programmes. People's understanding of this issue can regulate built-up development.

The leapfrog pattern develops due to the spatial heterogeneity of agricultural facilities [42], which is the case in the Batticaloa municipality area, such as Thiruperunthurai and Puthunagar. Thus, this discontinuous pattern forms the vacant land between built-up areas, making it difficult to afford facilities by the municipality. The transformation of non-urban land influences leapfrog and edge development into built-up land that increases faster than the growth of population in the Chinese cities [9]. However, this transformation was not identified in the Batticaloa municipality area during the selected periods. Rather, built-up discontinuous patterns were mainly identified in this city, which cause the rising cost of infrastructure.

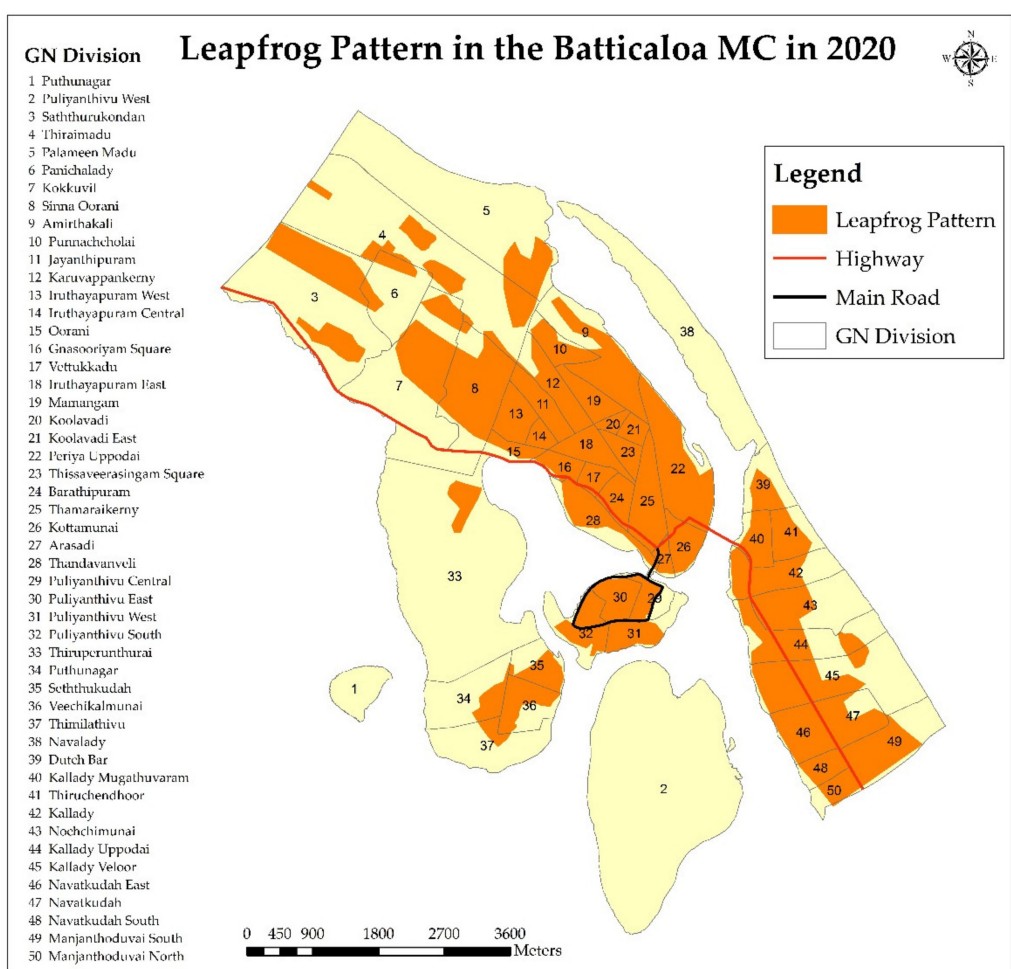

**Figure 8.** A leapfrog development pattern in the Batticaloa MC—2020.

### 4.2.3. Scattered Development

Scattered development also provides an inaccessible pattern on the urban edge, like undeveloped areas, creating sprawling. Figure 9 illustrates the scattered patterns in the study area that grows in the urban limit. The built-up growth develops in a dispersed way, which creates a considerable change in the city. This pattern was mainly identified in Thiruperunthurai and Navalady areas in the Batticaloa municipality. One reason for the scattered growth in the Thiruperunthurai area is that the land is mainly used for agriculture purposes. Municipal open spaces assigning a convenience value is one of the ways to account for scattered development. Therefore, people can be willing to spend more money on owning a home in these spacious areas, even if they are further away from the central business district, such as in the European cities [42]. This activity is sometimes experienced in Batticaloa city as well, in recent decades.

Further, the settlement areas of the core city developed by the individual homes in a vast land are mostly composed of single-storey buildings and some of double-storey houses. The individual housing preference of these people induces them to occupy the spacious land for building their dream house. Most families here are nuclear families rather than the extended family needed to build many individual houses for each family. For example, a nuclear family has four members, such as a father, mother, and two daughters. The parents must build two houses for these two daughters to marry them. Parents must build an individual house for each daughter, even if they have five daughters, because of the tradition in Batticaloa. A better solution for this, parents can build a low-rise building like three or four floors with all facilities and assign each floor to each daughter, which

can minimise the number of individual single homes in the city. This practice should be included in municipality policy, which makes the city more compact.

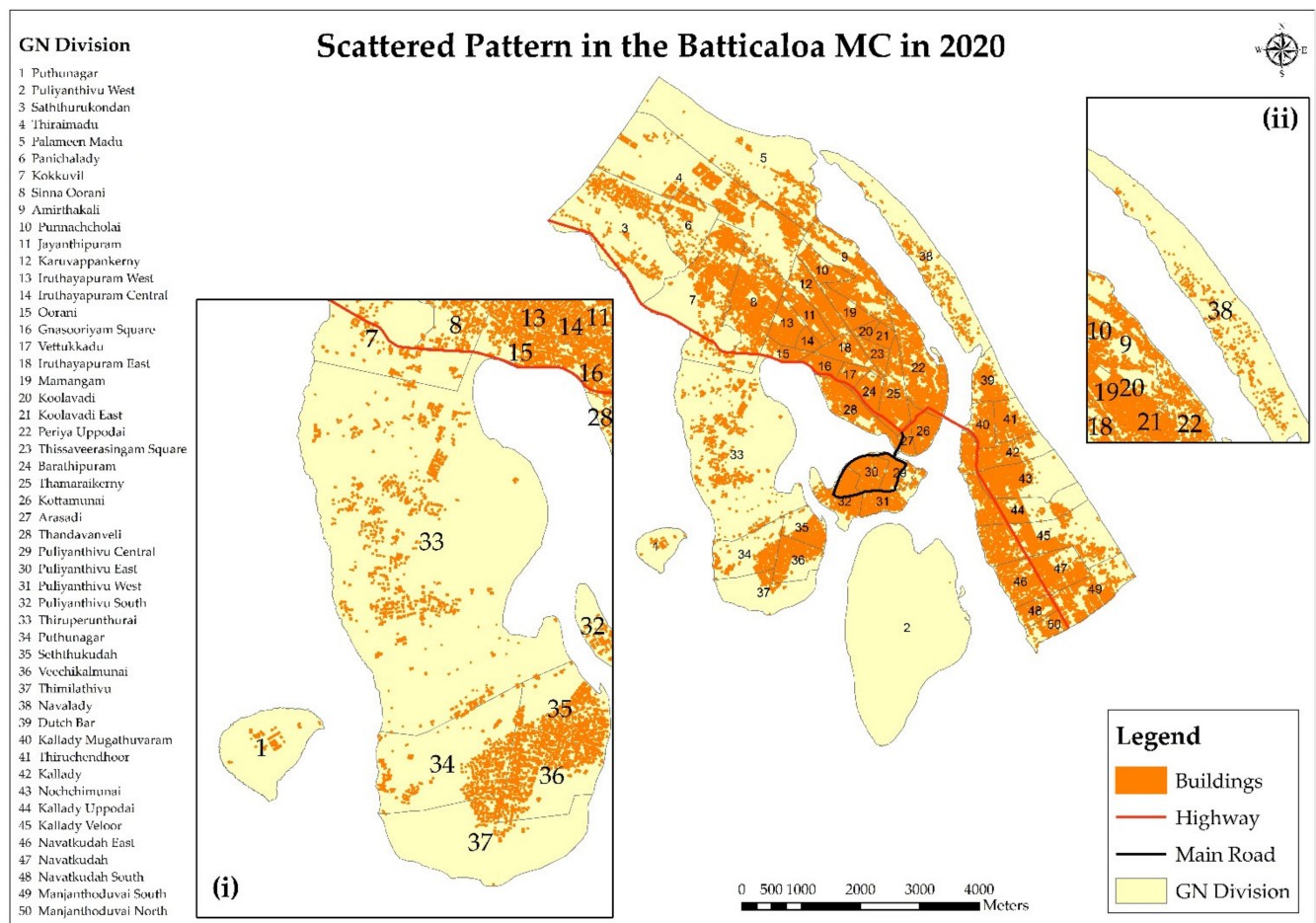

**Figure 9.** A scattered development pattern in the Batticaloa MC—2020, (**i**) and (**ii**) shows the scattered pattern in some GN divisions in the city.

### 4.2.4. Commercial Strip or Ribbon Development

Commercial development, along with highways, is another characteristic of urban sprawl called ribbon development that threatens sustainable urban growth in the city. Commercial buildings are mostly built along the main transport corridors in the core city and outside of the downtown area (see Figure 10). This type of development caused an increase in the value of the land near the highways. These urbanised areas use a mixed mode, such as commercial and residential, affecting urban land use. Many buildings utilise only the ground floor for commercial purposes, though in some buildings the second and third floors are also used as commercial spaces. Nevertheless, in the rest, all second or third levels are mainly used for residential purposes. People want to buy things such as clothes, grocery items, and electronic products for different purposes simultaneously, and they have to walk long distances to buy these things. This distance affects people's continuous shopping and time consumption.

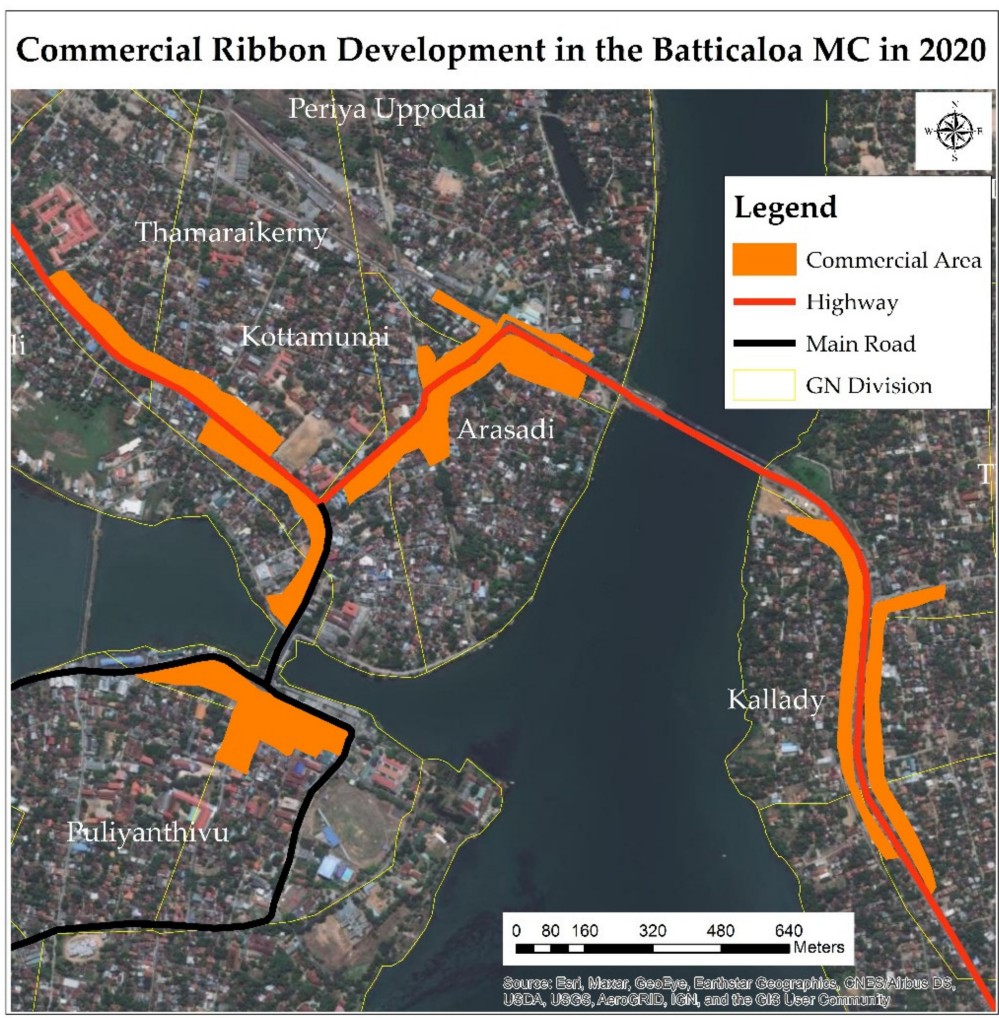

**Figure 10.** Commercial Ribbon Development in the Batticaloa MC—2020.

Further, this commercial ribbon pattern causes several problems in the city's functions. For example, approximately 75,000 to 100,000 people commute to Batticaloa municipality from 6 a.m. to 8 p.m. every day. The reasons for this commuting are to access services such as the railway station, the teaching hospital, the Faculty of Healthcare Sciences at the Eastern University, the Swami Vipulananda Institute of Aesthetic Studies at the Eastern University, the Open University, the district court, the airport, the technical college, the financial institutions, and other government institutions. Most of these institutions are located in commercial areas. Thus, this continuous commuting activity causes traffic congestion and frequent accidents in the city. Mainly Koddamaunai Bridge and the new bridge areas face traffic congestion daily. The main reason is that the traffic lights are not fixed in many areas of this city. Further, about 500 private buses and 320 public buses operate in the Batticaloa municipality area. This bus service starts from the Batticaloa's central bus station, which is located in the city centre, and has caused overcrowding. This ribbon development pattern is one of the main reasons for congestion in the city.

However, all these characteristics have created similar and different effects on the Batticaloa municipality. Leapfrog and scattered developments are the discontinuous and dispersed built-up growth that have less connectivity between the buildings. Meanwhile, commercial ribbon development with shopping complexes, restaurants, and banks built along the street of the core city generally depends on the highways for the developments. Various development projects were implemented in this area at the end of the civil war, which caused more changes in the city. Thus, land use classes were analysed to extract the area through characteristic changes in urban sprawl during the years 2000, 2010, and 2020.



The building patterns are not growing in a planned manner, such as housing, administration, shopping complexes, schools, police station, health office, and playgrounds. The police station and its quarters are built between the commercial area and also the core city. At the same time, administration buildings and playgrounds are also built in the core city. The suitability for the development of each sector was not considered until now. The main reason is less development planning in this area because it was affected by the civil war for three decades (1983–2009).

Further, the transportation network in the city was also not well designed. The primary and minor roads were not planned based on the network tracking method, which is more familiar for making road maps in ArcGIS. The road network of Batticaloa is not well mapped to identify the closest route between two locations in order to avoid traffic-related problems. For example, it is difficult to reach the general hospital of Batticaloa because of the lack of the closest facility in an emergency during traffic congestion. Closest facilities must consider tracking the places as quickly as possible to reach the location, and this must be a consideration when developing the transport pattern.

As a developing city in Sri Lanka, in Batticaloa the remaining categories that cause sprawl, such as auto-dependent or car-dependent development, uncontrolled growth, and uncoordinated growth, are not highly identified in this study area. The lower population of the city is the reason for these characteristics not having grown during the selected periods. Therefore, based on the identified characteristics of the study area, urban sprawl refers to the urban expansion [9,43] with low density beyond the built-up area [1,5,9,12,22,23], leapfrog development [5,12,13,26], a commercial ribbon development along the highways [12,13], and scattered growth [11,22,25].

*4.3. Future Plans and Regulations in the Batticaloa Municipality*

Plans and regulations regarding the construction of buildings, especially the construction of homes in current and future development, are the most important to minimise the urban sprawl development in the future. The development plan for Batticaloa municipality in 2030 is proposed in nine major zones, which are: residential zone, commercial zone, information technology zone, environmental conservation zone, airport-related activity zone, mixed development zone, administrative zone, fort conservation zone, and agricultural zone. These zones are separated by boundary lines such as roads or railways, or canals. These zones will only be used for the specific development for which the areas were designated. However, Batticaloa as a tourist city encourages activities related to tourism in any area of the municipality, depending on the suitability of the tourist development.

Further, the minimum land extent for the residential building is 6 perches, mentioned in the regulations of Batticaloa municipality. However, the maximum land area is not defined. Therefore, there are no obstacles to buying a large land and building a single house on this large land. Some people prefer the large spacious land to build their home today. The land for their preference is mostly available in the peripheral areas of the city. This development is the reason for the low density and scattered development patterns in the future as well.

Further, all buildings within 300 m of the coastal zone must be constructed with the prior approval of the Department of Coastal Conservation, Sri Lanka. A green belt was developed in the coastal strips to control the loss of biodiversity and the barriers from disasters such as tsunami and cyclone. Any building constructions in proximity to these disaster-prone areas should obtain clearance from the Urban Development Authority, Batticaloa. These regulations are the reasons for the low population and built-up density in the coastal areas of the city.

In addition, environmentally sensitive areas such as scrubland and mangrove forest are considered conservation areas, which do not allow the built-up development in the municipality areas, showing low density and leapfrog characteristics. Even though people can get approval to build houses in some environmental conservation areas, the buildings should be designed to retain natural beauty rather than obstruct the open green spaces.

Therefore, people should follow the regulations mentioned by the municipality to control the urban sprawl development.

However, Land Value Capture and Taxation are favourable for affordable housing development in the city with lower land value. Proper activation of this system can increase revenues and fix up downtown buildings. The significant increase in land value is due to the locational preferences as commercial areas and a transport hub. This tax system can control the increase of land value by the community preferences and use these tax revenues to develop public spaces such as bus stands, roads, and public spaces. Thus, the Municipality should strictly follow this policy in this city and encourage people to adopt this system to control land value, since people who pay tax can force the authorities to take care of their land and control the illegal occupancy of land in the city.

In addition, the urban development plan must also be open to feedback from stakeholders and experts who can make the development plan stronger. Most of the time, the municipality does not follow this practice when trying to implement a policy. The public is largely ignored in this process, which affects the sustainable planning of the city. Thus, the municipality should consider development activities in all aspects, including public participation. Therefore, proper regulations should be implemented based on the Sustainable Development Goal 11 and New Urban Agenda.

## 5. Conclusions

Urban sprawl has been increasing due to rapid construction development, especially housing in the Batticaloa municipal council. Urban fringe in the study area showed different sprawling characteristics, which are low density, leapfrog development, and scattered growth. However, a characteristic such as commercial ribbon development along the highways has been identified in the core city areas. Rapid population growth causes more sprawling development in the city, which was increasingly identified after 1990 in this area. In addition, people's preferences, land value, education, employment, income level, illegal construction, and permanent and temporary migrations are the main reasons for the sprawling development of the city. Based on these factors, it is clear that urban sprawl is a socioeconomic phenomenon that should be focused more on socioeconomic aspects in future studies.

Further, political influence interrupts the municipality's development plans and regulations. The rules and regulations need to be followed carefully in this area to develop itself as a sustainable city in the future. Maximum land extent for housing development should be stated in the regulations, which are the most important to minimise the pattern of low density and scattered development in the city. The municipality must identify existing illegal buildings, and when the buildings have not been built following the regulations, they must be demolished by the municipality. In addition, the illegal landfill in wetland areas such as scrubland areas must be punished and charged a penalty by the municipality. These regulations can control illegal activities in future. People have a lack of awareness of the urban sprawl development in this city; thus, it is important to educate them about the urban sprawl development and its effects through research, mapping, and community programmes.

Finally, this study identified the particular characteristics of urban sprawl such as low density, leapfrog development, scattered growth, and commercial ribbon development in the Batticaloa municipality. This finding empirically contributes to understanding the patterns of uncontrolled urban sprawl in Batticaloa city and other cities of Sri Lanka, and other developing countries in the future. This study can help to formulate strategic policies to minimise sprawling growth in the Batticaloa Municipal Council. Despite the above, this study, nevertheless, has limitations. Although the current low-resolution satellite images are deemed adequate in terms of accuracy, the findings could be enriched via a more high-resolution images analysis to identify micro-level changes and built-up patterns/forms. That may, in turn, provide more interesting explanations about urban sprawl characteristics. In addition, these sprawl characteristics creating impacts to land use/land cover patterns

are associated with physical and socioeconomic influences, but they are not considered in this study. Therefore, future studies should consider these influencing factors to obtain a holistic understanding and then form a predictive model curbing urban sprawl more effectively.

**Author Contributions:** Conceptualisation, methodology, M.S. and N.R.; software, M.S.; validation, M.S., N.R., G.H.T.L. and I.S.; formal analysis, M.S.; writing—original draft preparation, M.S.; writing—review and editing, M.S., N.R., G.H.T.L. and I.S.; supervision, N.R., G.H.T.L. and I.S.; funding acquisition, N.R. and G.H.T.L. All authors have read and agreed to the published version of the manuscript.

**Funding:** This research received no external funding.

**Institutional Review Board Statement:** Not applicable.

**Informed Consent Statement:** Not applicable.

**Acknowledgments:** We are very grateful to the Center for Innovative Planning and Development (CIPD) and the Faculty of Built Environment and Surveying, Universiti Teknologi Malaysia for the encouragement and financial support for this study, and our gratitude also goes to the reviewers who provided constructive comments and insights on this manuscript.

**Conflicts of Interest:** The authors declare no conflict of interest.

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
