# Peer review of "A Geo-Spatial Analysis for Characterising Urban Sprawl Patterns in the Batticaloa Municipal Council, Sri Lanka"

_land, doi:10.3390/land10060636_

Round 1

Reviewer 1 Report

This manuscript presents a clear and well-supported spatial analysis to characterize patterns of urban sprawl in Batticaloa municipality in Sri Lanka.  The maps presented are useful and convincing to present the results of the spatial analysis.The paper would benefit from a discussion of the existing and potential policy goals of the municipality in terms of urban sprawl, and the potential policy tools to cope with the effects of the urban sprawl identified.  The paper references the lack of road planning and general lack of spatial planning due to the conflict, but otherwise is thin in its discussion of the drivers of sprawl and the potential to manage it. It would be valuable to learn more about the potential policy responses to the urban sprawl identified and how the detailed spatial analysis in the paper could support such responses.

Author Response

Dear Reviewer,

Thank you very much for your comments and suggestions. We have revised the article based on the comments as follows,

  1. Plan and regulation, factors are explained (L353-361, L369-379, L387-394, L463-469, L481-486, L529-541, L646-659)
  2. Policy responses for urban sprawl addressed (L380-386, L395-401, L487-506, L570-574, L629-632)
  3. Future plans and regulations (L697-744)

Thank you.

Reviewer 2 Report

The articles provides interesting insights into urban sprawl in one city of Sri Lanka. The geodata anlysis is well prepared and presented.

However, there are serious weaknesses which have to be tackled before ublication:

  1. The research question which the article wants to address is not clearly formulated. It has to be based in the state of research and included in the article (abstract and main text).
  2. It is not at all clear why the case of Batticaloa is chosen and what it stands for. What can be learnt from the case study? What is new besides that there has not been a similar study on the city so far? Significance and relevance of the study? (e.g., L 14, 20, 57, 62, 522 ff).
  3. The study is very descriptive. Growth patterns are described. There is not much novelty in it. There are no considerations and explanations of reasons for the research. The property rights system and its relevance for urban sprawl is not even mentioned.
  4. The sentence regarding conclusions in the abstract (L28-31) is very unclear. The authors should clearly express their conclusions, including the novelty of the findings.
  5. There should be more recent data than the one used from 20213 (L38).
  6. The choice of Batticaloa seems very haphazard (L57). Why has this study been chosen? What is its significance for creating new knowledge? What can the readers learn especially from this case? How is the case related to the research question? Have the problems of a sigle case study been considered?
  7. Some sentences are not very clear, for example in L 77.
  8. The readability of Table 1 should be improved.
  9. The authors have to provide the definition which they follow (L70f). This is also important with regard to the (missing) research question and the methodology.
  10. Why was the study area chosen (L120ff)? Why and how were boundaries defined?
  11. The title of Figure 2 should be changed. I shows the land use (L151).
  12. The main body of the article contains nice maps but it is rather weak in explanations. No reasons for growth patterns are analysd and presented. As there is no research question, it is difficult to understand what the analysis is good for besides seeing it as another example for fact and trends which are well known and have been studied worldwide.
  13. The conclusions are very short (L522ff). They have to be related to a research question.
  14. Moreobver, the limitations of the study should be discussed.

Author Response

Reviewer 2 Comments:

The articles provides interesting insights into urban sprawl in one city of Sri Lanka. The geodata analysis is well prepared and presented.

However, there are serious weaknesses which have to be tackled before ublication:

  1. The research question which the article wants to address is not clearly formulated. It has to be based in the state of research and included in the article (abstract and main text).
  2. It is not at all clear why the case of Batticaloa is chosen and what it stands for. What can be learnt from the case study? What is new besides that there has not been a similar study on the city so far? Significance and relevance of the study? (e.g., L 14, 20, 57, 62, 522 ff).
  3. The study is very descriptive. Growth patterns are described. There is not much novelty in it. There are no considerations and explanations of reasons for the research. The property rights system and its relevance for urban sprawl is not even mentioned.
  4. The sentence regarding conclusions in the abstract (L28-31) is very unclear. The authors should clearly express their conclusions, including the novelty of the findings.
  5. There should be more recent data than the one used from 20213 (L38).
  6. The choice of Batticaloa seems very haphazard (L57). Why has this study been chosen? What is its significance for creating new knowledge? What can the readers learn especially from this case? How is the case related to the research question? Have the problems of a sigle case study been considered?
  7. Some sentences are not very clear, for example in L 77.
  8. The readability of Table 1 should be improved.
  9. The authors have to provide the definition which they follow (L70f). This is also important with regard to the (missing) research question and the methodology.
  10. Why was the study area chosen (L120ff)? Why and how were boundaries defined?
  11. The title of Figure 2 should be changed. I shows the land use (L151).
  12. The main body of the article contains nice maps but it is rather weak in explanations. No reasons for growth patterns are analysd and presented. As there is no research question, it is difficult to understand what the analysis is good for besides seeing it as another example for fact and trends which are well known and have been studied worldwide.
  13. The conclusions are very short (L522ff). They have to be related to a research question.
  14. Moreover, the limitations of the study should be discussed.

Dear Reviewer,

Thank you very much for your comments and suggestions. We have revised the article based on the comments as follows,

  1. Explanations of the reason for selected this area (L175-191),
  2. Its problem and question included (L18-20, L59-90)
  3. Significance included (L91-102)
  4. The novelty addressed (L769-773)
  5. Plan and property regulation, factors are explained (L353-361, L369-379, L387-394, L463-469, L481-486, L529-541, L646-659)
  6. Future plans and regulations (L697-744)
  7. The sentences clearly modified (L28-32, L110-118).
  8. The most recent data were used to explain the population growth (L38-41).
  9. Table 1 is adjusted in a single view (L131-133).
  10. Published past definitions were considered in the study to formulate the research question (L93-95, L131-133).
  11. Figure 2 title modified (L194-196).
  12. Discussion added for the results (Page 9-21 highlighted).
  13. Conclusion developed and added more based on research question (Page 21).
  14. The limitation of the study included (L774-783).

Thank you.

Round 2

Reviewer 2 Report

There are no further comments.